# WenXinGPT: A Multimodal Conversational Model for Enhancing Orthopedic Expert Consultations

## Abstract

Inspired by the hospital expert consultation model, this paper proposes a conversational medical visual language model for orthopedics, named WenXinGPT (Multi-disciplinary Collaboration). The core concept of this work focuses on aligning medical visual and textual representations to leverage high-quality data for generating expert consultation dialogues across hospital departments. The primary objective is to uncover orthopedic knowledge within medical intelligent models and enhance their reasoning abilities in an interpretable manner without requiring additional training. Our research particularly emphasizes zero-shot scenarios, and the results from experiments on 16 datasets provided by Peking Union Medical College Hospital demonstrate that the proposed WenXinGPT framework excels at mining and utilizing medical expertise within large language models, while also expanding their reasoning capabilities. Based on these findings, we conducted manual evaluations to identify and categorize common errors in our methods, along with ablation studies aimed at understanding the impact of various factors on overall performance.

## 1 Introduction

Large Language Models (LLMs) have revolutionized the field of natural language processing (NLP) by achieving unprecedented performance across a wide array of tasks (Radford et al., 2019; Brown et al., 2020; Chang et al., 2024). Models such as GPT-3 (Brown et al., 2020) and ChatGPT (Ouyang et al., 2022) have demonstrated remarkable capabilities in generating human-like responses, enabling advancements in machine translation, summarization, and conversational agents (Raffel et al., 2020). These models leverage deep neural architectures and vast datasets to capture complex linguistic patterns and semantic nuances (Vaswani et al., 2017; Liu et al., 2019), setting new benchmarks in NLP research.

The success of LLMs in textual domains has spurred interest in extending these models to handle multimodal data, integrating information from diverse sources such as text, images, and audio (Alayrac et al., 2022; Li et al., 2022b; Yang et al., 2024a). Multimodal models like MiniGPT-4 (Zhu et al., 2023), mPLUG-Owl (Ye et al., 2023), and LLaVA (Liu et al., 2023) have made significant strides by combining visual and textual information to perform tasks such as image captioning, visual question answering, and cross-modal retrieval (Li et al., 2023c; Gong et al., 2022). These models harness the synergy between different modalities to enhance understanding and generate more informative and context-aware responses.

Despite these advancements, the application of multimodal LLMs in specialized domains like medicine remains underexplored (Esteva et al., 2021; Miotto et al., 2018). The medical field poses unique challenges, including the need for high accuracy, interpretability, and compliance with ethical standards (Topol, 2019; Albahri et al., 2023). Medical data often consist of complex multimodal information, such as radiological images coupled with clinical notes, which require sophisticated models capable of integrating and interpreting heterogeneous data sources (Azizi et al., 2021; Schlemper et al., 2019). Moreover, there is a scarcity of large, high-quality medical datasets due to privacy concerns and regulatory constraints, hindering the training of robust models (Yang et al., 2019; Amiri et al., 2024).

Previous research on medical multimodal models, such as Visual-Med-Alpaca (Jiang et al., 2023), has made some progress by integrating medical image analysis with language models to assist in diagnostic processes. Similarly, studies have explored the use of deep learning for radiology report generation and medical image captioning (Jing et al., 2018; Li et al., 2018; Chen et al., 2020). However, these models primarily rely on English-language diagnostic reports, which presents a significant barrier to their adoption in non-English-speaking regions, particularly in Chinese medical contexts (Li et al., 2019; Zhou et al., 2021). This language barrier not only hinders research but also limits the practical deployment of such models in clinical settings where the local language is essential for effective communication (Wang et al., 2018; Huang et al., 2019). Additionally, current approaches often provide a single, final output without accommodating the iterative and interactive nature of medical decision-making processes (Kim et al., 2024; Wachter &

Cassel, 2021). The lack of multi-turn dialogue and intermediate reasoning steps reduces the utility of these models for healthcare professionals who rely on sequential and context-rich information (Gaube et al., 2021; Xie et al., 2020).

To address these challenges, we propose **WenXinGPT**, a novel multimodal large model tailored for orthopedic medical image diagnosis and interactive dialogue in Chinese. Our model is inspired by research in reinforcement learning and multi-agent systems (Silver et al., 2016; Vinyals et al., 2019), which enable agents to learn optimal policies through interaction with the environment and other agents. By incorporating these principles, WenXinGPT supports multi-round interactions that mirror real-world diagnostic processes, allowing for dynamic question-answering and decision support (Li et al., 2022a; Yu et al., 2019). The model not only provides intermediate outputs during the diagnostic dialogue but also integrates underlying prompt systems to guide the conversation and ensure a smoother workflow for healthcare professionals (Sun et al., 2021; Peng et al., 2022). This design contrasts with existing models, such as XrayGPT (Nguyen et al., 2022a), which often leave a substantial portion of interaction design to the user, making the systems less accessible to doctors unfamiliar with complex AI operations (Larasati et al., 2023; Van Berkel et al., 2023). Our contributions are as follows:

- We introduce **WenXinGPT**, a multimodal large model specifically designed for Chinese medical image diagnosis, addressing the lack of such models in non-English-speaking healthcare contexts.

- WenXinGPT integrates a multi-round interactive dialogue system, allowing for intermediate outputs that enhance decision support in clinical workflows.

- We implement an underlying prompt system that simplifies doctor-model interaction, making AI assistance more accessible and practical for medical professionals.

## 2 RELATED WORK

### 2.1 MULTIMODAL LARGE LANGUAGE MODELS

The integration of visual and linguistic modalities in large language models (LLMs) has garnered significant attention in recent years. To enable LLMs to process and comprehend visual information, various techniques have been developed. BLIP-2 (Li et al., 2023c) introduced Q-Former, which utilizes learnable query vectors to extract visual features from frozen image encoders, marking a notable advancement in aligning visual and language modalities. MiniGPT-4 (Zhu et al., 2023) demonstrated that further fine-tuning with detailed image descriptions can significantly improve the usability of multimodal models. LLAVA (Liu et al., 2023) extended this work by exploring diverse multimodal instruction-following data, including conversations, detailed descriptions, and complex reasoning tasks, with the goal of constructing a general-purpose visual assistant. Other notable contributions include Kosmos-2 (Peng et al., 2023) and VisionLLM (Chen et al., 2023), which focused on enhancing image comprehension, particularly in referring and grounding tasks.

Following the success of Image LLMs, researchers have shifted their focus to video comprehension, leading to the development of video-compatible LLMs. Notable examples include VideoChat (Li et al., 2023a), Video-LLaMA (Li et al., 2023b), and Video-ChatGPT (Zhou et al., 2023), which employ a two-stage training strategy. In the first stage, large-scale datasets are used to align video features with the LLM's feature space, and in the second stage, instruction tuning is performed using a limited set of annotated datasets. While these models have demonstrated impressive video comprehension abilities, their capabilities in temporal reasoning and describing specific video segments remain limited. This is largely due to the nature of datasets such as WebVid (Bain et al., 2021), which typically consist of one-event videos with noisy textual annotations. To address this limitation, our approach, WenXinGPT, introduces a fine-tuned interactive dialogue system that enhances the ability to handle detailed medical images and multiple rounds of dialogue in a clinical setting.

### 2.2 MEDICAL MULTIMODAL MODELS

In the medical field, multimodal models have also begun to emerge, though at a slower pace. Visual-Med-Alpaca (Jiang et al., 2023) is one of the few models designed to integrate medical images with diagnostic language models. This model, while promising, relies heavily on English diagnostic reports, limiting its application in non-English medical contexts, such as Chinese healthcare. Other notable efforts, such as XrayGPT (Nguyen et al., 2022a), have focused on generating reports from radiological images but lack the flexibility for multi-round interactions required in medical diagnoses. Additionally, these models often provide a single final output, which may not align well with iterative diagnostic processes typically used by healthcare professionals.

WenXinGPT builds on these foundations by specifically addressing the gap in Chinese-language medical multimodal models. Drawing inspiration from reinforcement learning and multi-agent systems (Silver et al., 2016; Vinyals et al., 2019), WenXinGPT introduces a multi-round interactive dialogue system that provides intermediate outputs, assisting doctors throughout the diagnostic process. Our model is tailored for medical image diagnosis in the orthopedic field and enhances doctor-patient interactions with dynamic dialogue and prompt systems. Unlike existing models, WenXinGPT offers not only diagnostic assistance but also a smoother, more intuitive interface for healthcare professionals, addressing the practical limitations of current medical multimodal systems.

## 3    DATASET

The dataset utilized in this study was obtained from Peking Union Medical College Hospital (PUMCH) and comprises 16 distinct and comprehensive data categories related to orthopedic surgery. All patient data have been desensitized to ensure confidentiality and privacy. Upon further processing and ethical approval, we intend to release the dataset publicly on GitHub to facilitate future research like (Wang et al., 2023; Yang et al., 2024b). The dataset includes both preoperative and postoperative medical records, imaging data, and patient rehabilitation details, providing a rich resource for training and evaluating medical models, particularly in the context of orthopedic surgery.

The key components of the dataset are as follows:

- **Patient Medical History:** This dataset captures comprehensive details regarding patients' health history, including chronic conditions, medication use, and allergies. This information plays a pivotal role in preoperative assessment and individualized surgical planning.

- **Surgical Objectives and Technique Selection:** Documentation of the specific surgical objectives for each case, along with the techniques and methods selected by the surgical team, provides valuable insights into clinical decision-making processes.

- **Neuroanatomical and Risk Assessment:** Neuroanatomical assessments, including risk evaluation, are conducted to identify potential challenges and risks associated with the surgery, particularly in relation to the patient's neurological condition.

- **Imaging Evaluations:** This dataset includes preoperative imaging such as CT scans and magnetic resonance imaging (MRI), which are essential for understanding the anatomical structure of the vertebral body and assessing the severity and location of fractures.

- **Postoperative Neurological Rehabilitation Plans:** Detailed rehabilitation plans developed post-surgery, outlining the recommended physical therapy regimens and neurological rehabilitation exercises, are provided to ensure optimal patient recovery.

- **Postoperative Neurological Function Records:** Neurological assessments conducted before and after surgery are recorded, including evaluations of sensation, motor function, and reflexes using standardized tools such as the Glasgow Coma Scale (GCS).

- **Surgical Records:** This component includes detailed records of the surgical procedures, such as the date and duration of surgery, the type of surgery performed, and the implants or devices used, offering valuable information for postoperative outcome analysis.

- **Postoperative Vital Signs Monitoring:** Continuous postoperative monitoring of key vital signs, including heart rate, blood pressure, and respiratory rate, to track the patient's recovery and identify any early signs of complications.

- **Neurological Function Assessments:** Records of neurological function pre- and post-surgery, focusing on sensory and motor function evaluations, provide critical insights into the patient's recovery trajectory.

- **Pain Assessments:** Pain levels are assessed using standardized tools to document the patient's pain experience over the course of recovery, along with the corresponding pain management interventions.

- **Local Trauma Observation:** This dataset includes observations of the surgical site postoperatively, documenting any complications such as bleeding, exudation, or signs of infection.

- **Postoperative Imaging:** Follow-up imaging data, such as X-rays and CT scans, are included to assess the outcomes of the surgery and the placement and integrity of any surgical implants or devices.

- **Rehabilitation Progress Records:** These records track the patient's progress through rehabilitation, documenting key milestones such as the initiation of physical therapy and the completion of various recovery stages.

- **Medication Records:** Documentation of postoperative medications prescribed, including antibiotics, anticoagulants, and analgesics, is included to monitor adherence to treatment and its effects on recovery.
- **Rehabilitation Plans and Recommendations:** Detailed rehabilitation plans tailored to each patient are included, with specific recommendations for physical therapy, exercises, and dietary adjustments to support recovery.
- **Complications and Adverse Reactions:** This dataset documents any complications or adverse events during the postoperative period, such as infections, bleeding, or nerve damage, which are critical for evaluating the safety and effectiveness of the surgical interventions.

The dataset provides a comprehensive longitudinal view of patient care, from preoperative assessments through postoperative recovery and rehabilitation. This breadth of data supports robust model development for various tasks, including surgical outcome prediction, rehabilitation monitoring, and postoperative complication detection. By including detailed records across multiple stages of the surgical process, this dataset represents a valuable resource for advancing the state of the art in medical multimodal modeling.

## 4 METHODOLOGY

In this section, we describe the methodology behind WenXinGPT, a 7-billion-parameter decoder-only language model optimized for medical applications, particularly in the domain of orthopedic surgery, as shown in Figure 1. The model incorporates advanced techniques such as Grouped-Query Attention (GQA), Neural Architecture Search (NAS), multi-department consultation (MC) modeling, and secondary pre-training on domain-specific medical data. The training process follows a structured pipeline involving pre-training, supervised fine-tuning, reward modeling, and reinforcement learning (RL).

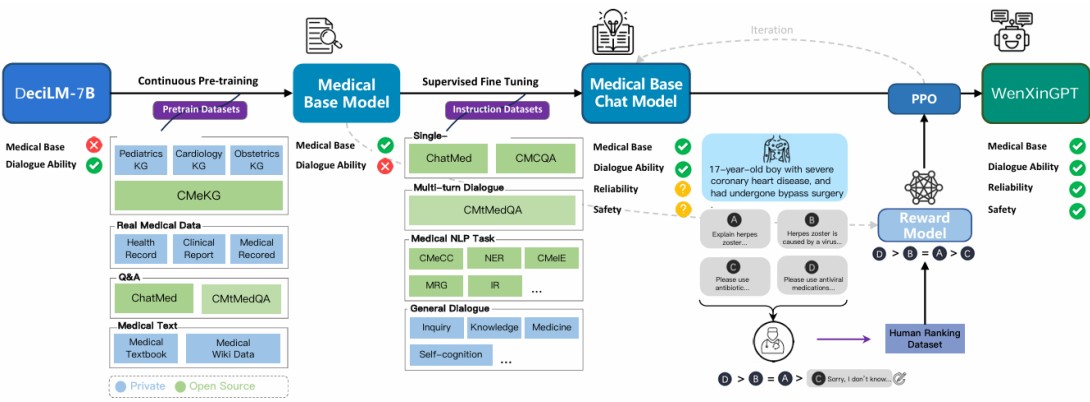

Figure 1: The overall fowchart of constructing WenXinGPT

### 4.1 MODEL ARCHITECTURE

WenXinGPT adopts a decoder-only architecture similar to the Transformer model (Vaswani et al., 2017), but with enhanced efficiency through the use of Grouped-Query Attention (GQA) (Shazeer, 2020). GQA reduces the computational complexity of self-attention by grouping multiple queries, keys, and values into smaller sets. Let the query, key, and value matrices for layer $l$ be denoted as $Q_l$, $K_l$, and $V_l$, respectively. The multi-head attention mechanism can be expressed as:

$$\text{Attention}(Q, K, V) = \text{softmax}\left(\frac{QK^T}{\sqrt{d_k}}\right)V \tag{1}$$

For Grouped-Query Attention, the attention mechanism is modified by processing the queries in groups, reducing computational overhead:

$$\text{GQA}(Q_l, K_l, V_l) = \sum_{g=1}^{G_l} \text{softmax}\left(\frac{Q_l^g (K_l^g)^T}{\sqrt{d_{k_g}}}\right)V_l^g \tag{2}$$

where $G_l$ is the number of groups in layer $l$, and $d_{k_g}$ is the dimensionality of the keys for group $g$. By grouping queries, GQA achieves a balance between accuracy and efficiency, enabling WenXinGPT to handle long sequences (up to 8K tokens).

## 4.2 Neural Architecture Search via AutoNAC

WenXinGPT's architecture is optimized using AutoNAC, a proprietary Neural Architecture Search (NAS) system from Deci (Deci, 2021). NAS automates the search for optimal network architectures by exploring a large space of candidate models and selecting the one that maximizes performance while minimizing computational cost. The optimization objective for AutoNAC can be formulated as a multi-objective loss function:

$$\mathcal{L}(\theta) = \mathcal{L}_t + \lambda_1 \mathcal{L}_c + \lambda_2 \mathcal{L}_{la} \tag{3}$$

where $\mathcal{L}_t$ represents task-specific losses (e.g., accuracy on medical diagnostics), $\mathcal{L}_c$ penalizes model complexity, and $\mathcal{L}_{la}$ accounts for inference latency. The parameters $\lambda_1$ and $\lambda_2$ control the trade-off between these objectives. AutoNAC performs an exhaustive search for the architecture $\theta^*$ that minimizes this loss:

$$\theta^* = \arg \min_\theta \mathcal{L}(\theta) \tag{4}$$

This search involves tuning hyperparameters such as the number of attention heads, hidden dimensions, and GQA group sizes, ensuring optimal performance on medical tasks.

## 4.3 Pre-training and Secondary Pre-training

WenXinGPT undergoes two main stages of training: general pre-training and secondary pre-training on medical data. The general pre-training stage is based on large-scale datasets, following the autoregressive language modeling objective:

$$\mathcal{L}_p = -\sum_{t=1}^{T} \log P(x_t | x_{<t}; \theta) \tag{5}$$

where $x_t$ represents the token at time step $t$, and $\theta$ are the model parameters. This pre-training equips the model with general linguistic capabilities.

In the secondary pre-training stage, WenXinGPT is further fine-tuned on domain-specific medical datasets, such as electronic health records (EHR) and imaging reports. The secondary pre-training follows a hybrid approach, combining supervised learning and reinforcement learning (RL). The supervised learning objective is defined as:

$$\mathcal{L}_s = -\sum_{i=1}^{N} \log P(y_i | x_i; \theta) \tag{6}$$

where $y_i$ is the correct label or output, and $x_i$ is the corresponding input. To refine the model's outputs in a medical context, we introduce reinforcement learning with human feedback (RLHF) (Ouyang et al., 2022), where the reward function $R(s)$ reflects the clinical relevance of the generated outputs:

$$\mathcal{L}_{RL} = -\mathbb{E}_{\pi_\theta} [R(s)] \tag{7}$$

where $\pi_\theta$ is the policy induced by the model parameters $\theta$, and $R(s)$ is the reward for the state $s$.

## 4.4 Multi-Department Consultation (MC) Framework

WenXinGPT integrates a novel Multi-Department Consultation (MC) framework to simulate real-world collaborative decision-making in healthcare. This approach models the interactions between medical experts from different departments (e.g., neurology, orthopedics, rehabilitation) who provide feedback on surgical plans. The multi-agent system can be formalized as a cooperative game, where each agent (expert) proposes a strategy $\pi_i$. The objective is to find a consensus strategy $\pi^*$ that maximizes the collective utility:

$$\pi^* = \arg \max_\pi \sum_{i=1}^{N} \alpha_i U(\pi, \pi_i) \tag{8}$$

where $U(\pi, \pi_i)$ is the utility of the final strategy $\pi$ relative to the expert's proposed strategy $\pi_i$, and $\alpha_i$ represents the weight of expert $i$'s opinion. This framework allows for transparent decision-making, where each department's suggestions are considered and evaluated. Pseudocode 1 outlines the key steps involved in the training of WenXinGPT:

---

**Algorithm 1** WenXinGPT Training Pipeline

---

1: **Input:** Pre-training data $\mathcal{D}_{\text{pretrain}}$, medical domain-specific data $\mathcal{D}_{\text{medical}}$, human feedback data $\mathcal{D}_{\text{feedback}}$, consultation data $\mathcal{D}_{\text{consultation}}$, learning rate $\eta$
2: **Output:** Final trained model $\theta^*$
3: Initialize model parameters $\theta_0$
4: **for** each mini-batch $\mathcal{B}$ in pre-training data $\mathcal{D}_{\text{pretrain}}$ **do**
5:     Compute pre-training loss $\mathcal{L}_{\text{p}}$ for batch $\mathcal{B}$
6:     Update model parameters:
$$\theta \leftarrow \theta - \eta \nabla_\theta \mathcal{L}_{\text{p}}$$
7: **end for**
8: **for** each mini-batch $\mathcal{B}$ in medical data $\mathcal{D}_{\text{medical}}$ **do**
9:     Compute supervised loss $\mathcal{L}_{\text{s}}$ for batch $\mathcal{B}$
10:     Update model parameters:
$$\theta \leftarrow \theta - \eta \nabla_\theta \mathcal{L}_{\text{s}}$$
11: **end for**
12: **for** each mini-batch $\mathcal{B}$ in human feedback data $\mathcal{D}_{\text{feedback}}$ **do**
13:     Compute reward $R(s)$ for generated output $s$
14:     Compute policy gradient loss $\mathcal{L}_{\text{RL}}$:
$$\mathcal{L}_{\text{RL}} = -\mathbb{E}_{\pi_\theta}\left[R(s)\right]$$
15:     Update model parameters:
$$\theta \leftarrow \theta - \eta \nabla_\theta \mathcal{L}_{\text{RL}}$$
16: **end for**
17: **for** each case $c$ in consultation data $\mathcal{D}_{\text{consultation}}$ **do**
18:     Collect expert opinions $\pi_1, \pi_2, \ldots, \pi_N$
19:     Aggregate expert plans into final plan $\pi^*$:
$$\pi^* = \arg\max_\pi \sum_{i=1}^{N} \alpha_i U(\pi, \pi_i)$$
20:     Output final surgical plan $\pi^*$
21: **end for**
22: **return** Final trained model $\theta^*$

---

The training of WenXinGPT follows a multi-objective optimization paradigm. The final objective combines the supervised learning loss $\mathcal{L}_{\text{s}}$, the reinforcement learning loss $\mathcal{L}_{\text{RL}}$, and the complexity constraints as follows:

$$\mathcal{L}_{\text{total}} = \mathcal{L}_{\text{s}} + \lambda_1 \mathcal{L}_{\text{RL}} + \lambda_2 \mathcal{L}_{\text{c}} \tag{9}$$

where $\lambda_1$ and $\lambda_2$ are hyperparameters balancing the different components. This allows for flexibility in fine-tuning the model while managing its computational overhead. The overall framework process of WenXinGPT is shown in Figure 2.

## 5 EXPERIMENTS

### 5.1 EXPERIMENT DETAILS

Our experimental setup for training WenXinGPT was divided into three distinct stages, each designed to enhance the model's capacity to generate high-quality medical reports and assist in multi-disciplinary expert consultation. These stages are described below:

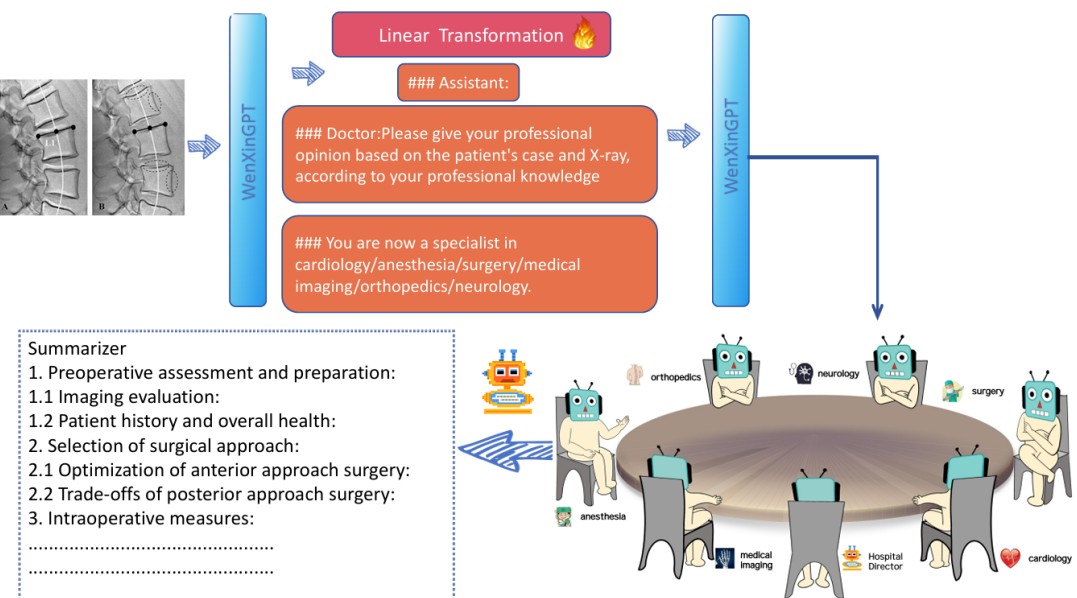

Figure 2: Overview of our WenXinGPT framework

Table 1: Model Architecture

| | |
|---|---|
| **Parameters** | 7.04 billion |
| **Layers** | 32 |
| **Heads** | 32 |
| **Sequence Length** | 8192 |
| **GQA num_key_value_heads** | Variable |

**Pre-training on Large-scale Medical Data**  In the first stage, we pre-trained the WenXinGPT model on a large corpus of electronic medical records and X-ray images. Specifically, we used 176,483 complete electronic medical records (EMRs) from high-quality orthopedic patients. The image data consisted of biplanar calibrated X-rays, captured using standard X-ray imaging devices, paired with their corresponding medical reports. The objective of this stage was to allow the model to learn the relationship between X-ray images and associated medical reports.

The model was trained using 16 A100 GPUs (32GB), with a total training step size of 320k and a batch size of 128. For this stage, we adapted the input data to the format used by the XrayGPT framework (Nguyen et al., 2022b). The rationale for employing this framework was to ensure that the data distribution was closely aligned with our model's target domain, optimizing data utilization.

**Fine-tuning with Domain-specific Data**  In the second stage, we fine-tuned the pre-trained model on a smaller, curated dataset of highly collated image-text pairs. These pairs were extracted from our internal dataset, which includes comprehensive EMRs and spinal X-ray images. The objective was to enhance the model's ability to generate natural, high-quality responses to spinal X-ray images.

We used 3,000 image-text pairs for this fine-tuning process, and the training was conducted on 2 A100 GPUs (32GB) with a batch size of 32 over 5,000 steps. This phase also continued leveraging the XrayGPT framework to further refine the model on more targeted, domain-specific data. Furthermore, Our model architecture is shown in the table 1.

**Multi-disciplinary Consultation and Report Summarization**  The third and final stage of training involved incorporating multi-disciplinary expert consultations into the WenXinGPT model. The model was tasked with simulating expert reasoning by analyzing and summarizing multiple rounds of discussions between medical professionals from various fields (orthopedics, neurology, cardiology, etc.).

Experts were asked to generate independent analyses, which were then aggregated by the model into comprehensive medical reports. The consultation process was modeled as a collaborative decision-making framework, where each expert provided input, voted on the proposed summaries (yes/no), and suggested improvements. The model iteratively refined the reports based on expert feedback.

We also compared the performance of WenXinGPT with GPT-3.5 and XrayGPT frameworks across five experimental setups: Zero-shot, Zero-shot CoT (Chain of Thought), Few-shot, Few-shot CoT, and Few-shot CoT + SC (Self-consistency) (Kojima et al., 2022; Wei et al., 2022). In all experiments, temperature and top-p were set to 1.0. A total of 200 samples per dataset were randomly selected for the final evaluation, with prompts available on our Github repository.

Table 2: Comparison of our WenXinGPT with Baseline using the Rogue score on the Electronic medical record for orthopedic patients at Peking Union Medical College Hospital.The best result is shown in blue, the second best in pink, and the third best in a lighter shade of pink.

| Method | R-1 Score | R-2 Score | R-L Score |
|---|---|---|---|
| **GPT-3.5** | | | |
| **- Zero-shot** | 0.4177 | 0.0208 | 0.0329 |
| **- Zero-shot CoT** | 0.4216 | 0.0328 | 0.0448 |
| **- Few-shot** | 0.4595 | 0.0533 | 0.1342 |
| **- Few-shot CoT** | 0.4659 | 0.0756 | 0.1556 |
| **- Few-shot CoT + SC** | 0.4783 | 0.0969 | 0.1967 |
| **XrayGPT** | | | |
| **- Zero-shot** | 0.5193 | 0.0578 | 0.0934 |
| **- Zero-shot CoT** | 0.5416 | 0.0649 | 0.1056 |
| **- Few-shot** | 0.5898 | 0.0845 | 0.1442 |
| **- Few-shot CoT** | 0.6212 | 0.0974 | 0.1892 |
| **- Few-shot CoT + SC** | 0.6381 | 0.1376 | 0.2038 |
| **WenXinGPT** | | | |
| **- Zero-shot CoT** | 0.6293 | 0.1629 | 0.2594 |
| **- Zero-shot CoT** | 0.6484 | 0.1792 | 0.2606 |
| **- Few-shot** | 0.6532 | 0.1932 | 0.2793 |
| **- Few-shot CoT** | 0.6738 | 0.2144 | 0.2985 |
| **- Few-shot CoT + SC** | 0.6922 | 0.2389 | 0.3244 |

## 5.2 Evaluation Metrics

We evaluated the performance of WenXinGPT and the baselines using multiple advanced metrics. The primary metric for text generation quality was the ROUGE score (Lin, 2004), which measures the overlap between generated text and reference summaries. Specifically, we used ROUGE-L, which emphasizes the longest common subsequence between generated reports and reference texts, offering insight into content similarity.

Additionally, we employed GPT-based evaluations (Zhong et al., 2023), where a GPT model is used to judge the fluency, coherence, and factual correctness of generated outputs compared to reference reports.

## 5.3 Quantitative Results

Our quantitative results, shown in Table 2, highlight the performance advantages of WenXinGPT over the baseline models. WenXinGPT achieved a 39% improvement over the state-of-the-art XrayGPT framework on the test set from Peking Union Medical College Hospital, which included orthopedic medical image-electronic medical record pairs. This significant improvement demonstrates WenXinGPT's effectiveness in generating accurate, detailed medical reports for orthopedic surgery.

## 5.4 WenXinGPT Example Case

In this section, we present a detailed case study to demonstrate the practical applicability of WenXinGPT in a complex medical scenario. The subject is a 17-year-old male patient diagnosed with severe coronary heart disease, who had previously undergone coronary artery bypass surgery. A recent computed tomography (CT) scan revealed a fracture in the C7 vertebra of the cervical spine. Given the complexity of the patient's medical history and the need for interdisciplinary expertise, a multi-department expert consultation was convened to design an optimal treatment plan.

The consultation included specialists from orthopedics, surgery, imaging, anesthesiology, neurology, and cardiovascular medicine as shown in Figure 3.

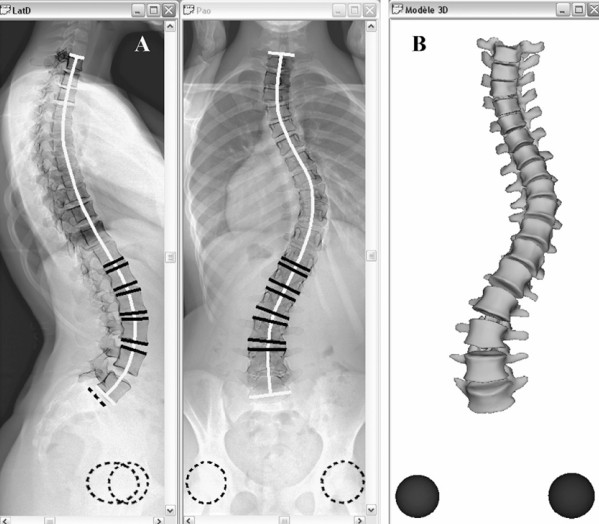

Figure 3: (A) The digitization of several descriptor parameters (white parameters) of the parameter spine allows estimation of other descriptors (black parameters) using vertical infer ence. The estimate of the 3D reconstruction (B) is projected on the biplane X-ray (A). The digitization of the pelvis is in dicated by the black dotted line (A)

**Orthopedic Recommendations:** The orthopedic team proposed a posterior surgical approach involving pedicle screw fixation of the left lateral mass of the atlas. This conservative approach is preferred for stabilizing the vertebra, particularly given the fragility of the cervical spine in such cases (Hartenberger et al., 2020). In addition, an anterior approach was considered as a viable alternative for addressing the C7 vertebral fracture. However, the anterior approach, despite providing direct access to the vertebral body, carries the risk of exacerbating stress points, which could lead to further complications during surgery.

**Surgical Considerations:** The surgical team concurred with the use of the posterior approach, recommending internal fixation followed by postoperative external fixation with cervical support. The team also assessed the potential risks associated with the anterior approach, emphasizing the possibility of fragment displacement into the spinal canal during resection, which could lead to spinal cord compression and catastrophic neurological outcomes (Smith et al., 2018). They highlighted that employing blunt dissection could reduce clotting-related complications, although this might extend the surgical duration.

**Cardiovascular Assessment:** The cardiovascular team underscored the importance of protecting the vascular structures in the cervical region, especially given the instability of the spine caused by the C7 fracture. They expressed concerns that vertebral fractures could disrupt the arteries supplying blood to the cervical spine, potentially leading to ischemia in the vertebral and surrounding tissues (Moore et al., 2021). The team recommended adjusting the patient's anticoagulation therapy, particularly if the patient was receiving medications like warfarin, and suggested bridging therapy with low molecular weight heparin (LMWH) to maintain optimal anticoagulation status during surgery.

**Anesthesiology Input:** The anesthesiology team discussed the risks of anticoagulation management, noting that improper dosing of anticoagulants such as warfarin or heparin could significantly increase the risk of intraoperative bleeding. They also cautioned that overuse of clotting factor supplementation could trigger abnormal bleeding due to excessive thrombinolytic activation (Haug et al., 2020). Given these risks, they recommended that any clotting protocol be tailored carefully to the patient's specific coagulation profile.

**Imaging and Neurological Risks:** Imaging and neurology specialists focused on the potential for vertebral artery rupture, a complication that could lead to reduced blood supply to the spinal cord, causing ischemia and further neurological damage. They highlighted that preserving the integrity of the vertebral artery is critical to preventing spinal cord infarction and subsequent neurological deficits (Morrison et al., 2017).

**Final Surgical Plan:** Based on the input from the interdisciplinary team, the following surgical plan was developed:

- **Posterior Approach Surgery:** The primary recommendation is to perform posterior approach surgery with pedicle screw fixation to stabilize the C7 vertebra. This conservative approach minimizes interference with surrounding structures and reduces surgical risks.
- **Increased Traction:** Additional traction on the cervical spine will be applied to ensure stability during the procedure, mitigating the risk of further vertebral displacement.
- **Blunt Dissection:** Blunt dissection will be used to open the surgical field, minimizing disruption to the clotting system and reducing the likelihood of excessive bleeding.

**Cardiovascular Considerations:**

- **Vascular Protection:** Special precautions will be taken to protect the vascular structures, particularly the nourishing arteries in the cervical region, to prevent any disruption of blood flow to the spinal cord and surrounding tissues.
- **Pre-coagulation Protocol:** If pre-coagulation treatment is deemed necessary, anticoagulation therapy will be carefully managed, potentially including the adjustment or discontinuation of medications like warfarin, with bridging therapy provided where needed.

**Surgical Risks:**

- **Vertebral Artery Rupture:** The risk of vertebral artery rupture will be closely monitored, as this complication could result in severe spinal cord ischemia and neurological impairment.
- **Anterior Approach Considerations:** While the anterior approach offers direct access to the vertebral body, it carries significantly higher risks due to the proximity of critical structures such as the trachea, esophagus, and major blood vessels. Therefore, this approach will be used only if absolutely necessary.
- **Extended Surgical Time:** The use of blunt dissection may prolong the surgery, but the reduced risk of bleeding justifies the extended operative time.

Following extensive consultation, the consensus among the interdisciplinary team is that a posterior approach surgery with pedicle screw fixation should be prioritized. This approach is the least invasive and offers the best balance between stabilizing the vertebrae and minimizing the risk of complications, particularly given the patient's severe coronary heart disease and complex medical history.

WenXinGPT was instrumental in synthesizing the diverse expert inputs into a cohesive and comprehensive surgical plan, demonstrating its potential utility in multi-disciplinary clinical decision-making. This case exemplifies the critical role of AI-driven systems like WenXinGPT in supporting healthcare professionals by integrating large-scale medical knowledge with real-time expert analysis to improve patient outcomes.

## 6 CONCLUSION

In this study, we introduced WenXinGPT, a large language model designed to enhance medical decision-making through multi-disciplinary expert consultations. Trained on large-scale electronic medical records and X-ray images, and fine-tuned with domain-specific data, WenXinGPT significantly outperformed existing models like GPT-3.5 and XrayGPT in generating accurate and coherent medical reports. The model's application to a complex orthopedic surgery case demonstrated its ability to synthesize expert inputs across various specialties. WenXinGPT represents a promising advancement in AI-assisted healthcare, offering a robust decision-support tool capable of improving patient outcomes and streamlining expert collaboration in clinical settings.

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

## A    LIMITATIONS

The validity and applicability of the WenXinGPT model are currently mainly evaluated on Chinese medical data. The generalization of the model to other language and medical Settings remains an unexplored area. WenXinGPT's success depends on the availability of multi-modal medical data, particularly in the orthopaedic field. There is no explicit proposal to extend it to other medical specialties. Such as heart disease, neurological disease and so on. While WenXinGPT aims to improve reasoning in an explainable way, there can be challenges in fully understanding and interpreting the model's decisions, especially in complex medical scenarios. Ensuring transparency and interpretability remains an ongoing concern. This paper explores the potential of "WenXinGPT" to assist doctors in case analysis, but the practical application of this model to the clinic may face resistance or challenges from the medical community, which needs to be further explored and verified. Although manual evaluation and ablation studies are mentioned in this paper to evaluate the performance of the model, there may be limitations in the selected evaluation measures Exploring a wider range of indicators and considering real-world clinical outcomes can enhance the robustness of the assessment. Ethical issues in integrating AI models into healthcare should be thoroughly addressed, including issues related to patient privacy, data security and model bias.

