# OpenReview forum: "WenXinGPT: A Multimodal Conversational Model for Enhancing Orthopedic Expert Consultations"
_ICLR.cc/2025/Conference — ICLR 2025 Conference Withdrawn Submission_

### Official Review · Reviewer_QMrA · 2024-10-18

**Soundness:** 2
**Presentation:** 1
**Contribution:** 2
**Rating:** 3
**Confidence:** 5

**Summary:**

This paper introduces WenXinGPT, which incorporates multiple LLM agents on X-ray images for better clinical diagnosis. The idea of using multi-agents is interesting. However, there are major flaws that I will outline in more detail.

**Strengths:**

1. The use of multi-agents. This design makes the generation of diagnosis more like a 'joint expert consultation' process, improving the outputs' robustness and interoperability.
2. The generation of the dataset. The dataset mentioned in the paper, if publically available, would be a good platform for future research.

**Weaknesses:**

1. The paper is not well written.
    1) For example, the authors mentioned in the contributions, 'implement an underlying prompt system'. However, this part is missing in the paper.
     2) The dataset is not clearly introduced. How many records are in this dataset? How many participants are in this dataset?
     3) How are the D_consultations (mentioned in the Training Pipeline) acquired? And how is the human feedback acquired for RLHF?

2. Some of the contents are misleading. For example, the authors mentioned that they use a 'a 7-billion-parameter decoder-only LM', which turns out to be DeciLM-7B developed by others. Did the authors make modifications? Why don't citation the DeciLM-7B at the first time it appears? Did the authors develop GQA, NAS? Or just use the implementation the same as Deci? This needs to be clarified.

3. The experimental design is not clear. For quantification evaluation (testing), which portion of data was used? How is the performance of BLEU scores? How are CoT, SC, and few-shot, zero-shot strategies implemented? Why just compare with GPT-3.5 and XtayGPT, instead of other general LLMs and medical LLMs? With Few-shot CoT + SC, the performance is better than WenXinGPT itself. How to further improve the performance of WenXinGPT? How is the 'consensus among the interdisciplinary team' reached in the example case?

**Questions:**

I have strong concerns regarding to the three contributions that the authors mentioned:
1. The authors mentioned the first contribution is to fill the gap of non-English-speaking healthcare LLMs. If so, why don't we just translate the existing English-based LLMs model to the target language? Will that lead to a decreased performance?
2. Does 'multi-round interactive dialogue system' refer to the multi-agents? This should be more like a 'joint expert consultation' process rather than a 'multi-round interactive dialogue'. How is the  'consensus among the interdisciplinary team' reached?
3. How is the 'underlying prompt system' involved in this research? This part is missing.

**Details Of Ethics Concerns:**

No.

---

### Official Review · Reviewer_4LpN · 2024-11-03

**Soundness:** 2
**Presentation:** 2
**Contribution:** 3
**Rating:** 3
**Confidence:** 4

**Summary:**

This paper introduces WenXinGPT, a multimodal LLM for orthopedic medical diagnoses in Chinese. This paper introduces a new dataset for orthopedic surgery and uses a Multi-Department Consultation framework to develop a comprehensive surgical plan.

**Strengths:**

1. This paper addresses a significant gap in non-English healthcare by introducing a multimodal orthopedic domain language model in Chinese.
2. Introduced a novel MC approach that includes feedback from various experts in formalizing the final surgical plan.
3. Incorporates multi-round discussion amongst medical professionals from different domains, thus aligning it closely with real-world medical consultations.
4. Introduced a new dataset containing detailed categories of orthopedic surgery essential for future research in this domain.

**Weaknesses:**

1. Dataset details: Details on dataset size (number of tokens), high level statistical analysis, and dataset composition are lacking, including the specific datasets used and the proportions allocated for pretraining and fine-tuning (SFT).
2. Evaluation Metrics: Evaluation relies solely on ROUGE scores, which is insufficient to capture essential aspects of medical report quality, such as interpretability and usability. Comparisons are limited to two other LLMs; additional comparisons to advanced models like Opus or GPT-4 would better contextualize the results. Results from GPT-based assessments also need to be included. The work will significantly benefit from human evaluations.
3. Ablation Studies: The study needs an analysis of how NAS and MC strategies impact model performance, making the effectiveness of these approaches unclear.
4. Implementation Details: Key implementation details, such as the prompts used for the MC framework and the tasks for supervised fine-tuning (SFT), must be included, impacting reproducibility.
5. Multi-Turn Dialogue: The multi-turn interaction mechanism needs to be clearly explained, and the example provided needs to illustrate how multi-turn discussions are initiated or maintained sufficiently.
6. Domain Focus and Generalizability: The choice to focus exclusively on orthopedics is not entirely justified, and there is limited discussion on the model’s adaptability to other medical specialties or non-Chinese datasets.
7. Ethical Considerations: Information on handling Protected Health Information (PHI) in the dataset is incomplete, with no clear explanation of the PHI removal or validation techniques.

**Questions:**

Comments: This paper represents valuable groundwork for healthcare applications in non-English languages, and I believe it addresses an important and necessary area. However, it currently lacks significant details that require attention.
Suggestions:
Please address the points outlined in the Weakness section to enhance the paper's contribution. Specifically, expanding on evaluation metrics and experiments, ablation work to showcase the impact of MC, as well as expanding on the multi-turn approach, would greatly strengthen the technical contribution of this paper.

**Details Of Ethics Concerns:**

The authors state that all patient data has been desensitized to protect confidentiality and privacy; however, they do not provide further details or evidence to substantiate this claim. Hence, an ethical review is needed.

---

### Official Review · Reviewer_47Ub · 2024-11-06

**Soundness:** 1
**Presentation:** 2
**Contribution:** 1
**Rating:** 3
**Confidence:** 4

**Summary:**

This paper introduces WenXinGPT, a 7B parameter multimodal language model designed for orthopedic medical consultations in Chinese healthcare settings. The authors present a three-stage training process involving pretraining, domain-specific fine-tuning, and incorporation of multi-disciplinary expert consultations. The model is evaluated on medical data and compared against GPT-3.5 and XrayGPT using ROUGE.

**Strengths:**

The authors create a comprehensive dataset covering 16 distinct categories of orthopedic surgery-related data from a medical institution. The dataset includes diverse medical information could be valuable for future research.

**Weaknesses:**

The paper claims to be multimodal, stating "WenXinGPT, a multimodal large model specifically designed for Chinese medical image diagnosis" (Introduction), yet provides no technical details about visual processing or multimodal integration architecture.

The evaluation is fundamentally flawed, comparing a supposedly multimodal model primarily against GPT-3.5 (a text-only model) using only text-based ROUGE metrics. Despite citing MiniGPT-4, LLaVA, and mPLUG-Owl in the introduction, these more relevant multimodal baselines are absent from the evaluation.

The paper claims architectural innovations through NAS and GQA but provides no evidence these choices improve upon existing architectures like Llama that already use GQA. Testing is limited to a single institutional dataset, raising questions about generalizability.

**Questions:**

1. Can you provide results on other medical datasets to demonstrate generalizability?

2. Why were modern multimodal models not included as baselines? The current comparison against GPT-3.5 seems inappropriate for evaluating multimodal capabilities.

3. The paper mentions using "16 A100 GPUs (32GB)" for training, but A100s only come in 40GB and 80GB variants. Could you what models were used?

4. What specific advantages does your architecture provide over existing models like Llama 3 that already use GQA?

---

### Note · Authors · 2024-11-15

**Comment:**

The suggestions made by the reviewers have helped us make our article better. We are willing to listen to the reviewers' opinions and make long revisions, which may not be in time for the next round of reviews. Thank you for the reviewers' fair evaluation of our article.

**Withdrawal Confirmation:**

I have read and agree with the venue's withdrawal policy on behalf of myself and my co-authors.